# IMPROVING DEEP REGRESSION WITH ORDINAL ENTROPY

**Shihao Zhang[1], Linlin Yang[1], Michael Bi Mi[2], Xiaoxu Zheng[2], Angela Yao[1]**
[1]National University of Singapore, Singapore,
[2]Huawei International Pte Ltd, Singapore

## ABSTRACT

In computer vision, it is often observed that formulating regression problems as a classification task yields better performance. We investigate this curious phenomenon and provide a derivation to show that classification, with the cross-entropy loss, outperforms regression with a mean squared error loss in its ability to learn high-entropy feature representations. Based on the analysis, we propose an ordinal entropy regularizer to encourage higher-entropy feature spaces while maintaining ordinal relationships to improve the performance of regression tasks. Experiments on synthetic and real-world regression tasks demonstrate the importance and benefits of increasing entropy for regression. Code can be found here: https://github.com/needylove/OrdinalEntropy

## 1 INTRODUCTION

Classification and regression are two fundamental tasks of machine learning. The choice between the two usually depends on the categorical or continuous nature of the target output. Curiously, in computer vision, specifically with deep learning, it is often preferable to solve regression-type problems as classification tasks. A simple and common way is to discretize the continuous labels; each bin is then treated as a class. After converting regression into classification, the ordinal information of the target space is lost. Discretization errors are also introduced to the targets. Yet for a diverse set of regression problems, including depth estimation (Cao et al., 2017), age estimation (Rothe et al., 2015), crowd-counting (Liu et al., 2019a) and keypoint detection (Li et al., 2022), classification yields better performance.

The phenomenon of classification outperforming regression on inherently continuous estimation tasks naturally begs the question of why. Previous works have not investigated the cause, although they hint at task-specific reasons. For depth estimation, both Cao et al. (2017) and Fu et al. (2018) postulate that it is easier to estimate a quantized range of depth values rather than one precise depth value. For crowd counting, regression suffers from inaccurately generated target values (Xiong & Yao, 2022). Discretization helps alleviate some of the imprecision. For pose estimation, classification allows for the denser and more effective heatmap-based supervision (Zhang et al., 2020; Gu et al., 2021; 2022).

Could the performance advantages of classification run deeper than task-specific nuances? In this work, we posit that regression lags in its ability to learn high-entropy feature representations. We arrive at this conclusion by analyzing the differences between classification and regression from a mutual information perspective. According to Shwartz-Ziv & Tishby (2017), deep neural networks during learning aim to maximize the mutual information between the learned representation $Z$ and the target $Y$. The mutual information between the two can be defined as $\mathcal{I}(Z;Y) = \mathcal{H}(Z) - \mathcal{H}(Z|Y)$. $\mathcal{I}(Z;Y)$ is large when the marginal entropy $\mathcal{H}(Z)$ is high, *i.e.*, features $Z$ are as spread as possible, and the conditional entropy $\mathcal{H}(Z|Y)$ is low, *i.e.*, features of common targets are as close as possible. Classification accomplishes both objectives (Boudiaf et al., 2020). This work, as a key contribution, shows through derivation that regression minimizes $\mathcal{H}(Z|Y)$ but ignores $\mathcal{H}(Z)$. Accordingly, the learned representations $Z$ from regression have a lower marginal entropy (see Fig. 1(a)). A t-SNE visualization of the features (see Fig. 1(b) and 1(c)) confirms that features learned by classification have more spread than features learned by regression. More visualizations are shown in Appendix B.

The difference in entropy between classification and regression stems from the different losses. We postulate that the lower entropy features learned by $L_2$ losses in regression explain the performance

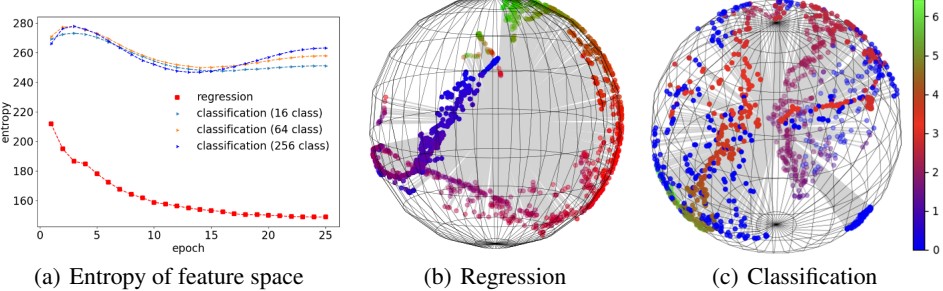

| (a) Entropy of feature space | (b) Regression | (c) Classification |

Figure 1: Feature learning of regression versus classification for depth estimation. Regression keeps features close together and forms an ordinal relationship, while classification spreads the features (compare (b) vs. (c) ), leading to a higher entropy feature space. Features are colored based on their predicted depth. Detailed experimental settings are given in Appendix B.

gap compared to classification. Despite its overall performance advantages, classification lags in the ability to capture ordinal relationships. As such, simply spreading the features for regression to emulate classification will break the inherent ordinality of the regression target output.

To retain the benefits of both high entropy and ordinality for feature learning, we propose, as a second contribution, an ordinal entropy regularizer for regression. Specifically, we capture ordinal relationships as a weighting based on the distances between samples in both the representation and target space. Our ordinal entropy regularizer increases the distances between representations, while weighting the distances to preserve the ordinal relationship.

The experiments on various regression tasks demonstrate the effectiveness of our proposed method. Our main contributions are three-fold:

- To our best knowledge, we are the first to analyze regression's reformulation as a classification problem, especially in the view of representation learning. We find that regression lags in its ability to learn high-entropy features, which in turn leads to the lower mutual information between the learned representation and the target output.

- Based on our theoretical analysis, we design an ordinal entropy regularizer to learn high-entropy feature representations that preserve ordinality.

- Benefiting from our ordinal entropy loss, our methods achieve significant improvement on synthetic datasets for solving ODEs and stochastic PDEs as well as real-world regression tasks including depth estimation, crowd outing and age estimation.

## 2 RELATED WORK

**Classification for Continuous Targets.** Several works formulate regression problems as classification tasks to improve performance. They focus on different design aspects such as label discretization and uncertainty modeling. To discretize the labels, Cao et al. (2017); Fu et al. (2018) and Liu et al. (2019a) convert the continuous values into discrete intervals with a pre-defined interval width. To improve class flexibility, Bhat et al. (2021) followed up with an adaptive bin-width estimator. Due to inaccurate or imprecise regression targets, several works have explored modeling the uncertainty of labels with classification. Liu et al. (2019a) proposed estimating targets that fall within a certain interval with high confidence. Tompson et al. (2014) and Newell et al. (2016) propose modeling the uncertainty by using a heatmap target in which each pixel represents the probability of that pixel being the target class. This work, instead of focusing on task-specific designs, explores the difference between classification and regression from a learning representation point of view. By analyzing mutual information, we reveal a previously underestimated impact of high-entropy feature spaces.

**Ordinal Classification.** Ordinal classification aims to predict ordinal target outputs. Many works exploit the distances between labels (Castagnos et al., 2022; Polat et al., 2022; Gong et al., 2022) to

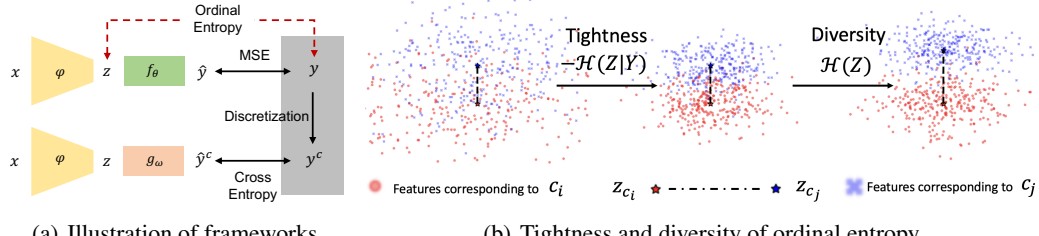

(a) Illustration of frameworks        (b) Tightness and diversity of ordinal entropy

Figure 2: Illustration of (a) regression and classification for continuous targets, and the use of our ordinal entropy for regression, (b) the pull and push objective of tightness and diversity on the feature space. The tightness part encourages features to be close to their feature centers while the diversity part encourages feature centers to be far away from each other.

preserve ordinality. Our ordinal entropy regularizer also preserves the ordinality by exploiting the label distances, while it mainly aims at encouraging a higher-entropy feature space.

**Entropy.** The entropy of a random variable reflects its uncertainty and can be used to analyze and regularize a feature space. With entropy analysis, Boudiaf et al. (2020) has shown the benefits of the cross-entropy loss, *i.e.*, encouraging features to be dispersed while keeping intra-class features compact. Moreover, existing works (Pereyra et al., 2017; Dubey et al., 2017) have shown that many regularization terms, like confidence penalization (Pereyra et al., 2017) and label smoothing (Müller et al., 2019), are actually regularizing the entropy of the output distribution. Inspired by these works, we explore the difference in entropy between classification and regression. Based on our entropy analysis, we design an entropy term (*i.e.*, ordinal entropy) for regression and bypass explicit classification reformulation with label discretization.

## 3    A Mutual-Information Based Comparison on Feature Learning

### 3.1    Preliminaries

Suppose we have a dataset $\{X, Y\}$ with $N$ input data $X = \{\mathbf{x}_i\}_{i=1}^N$ and their corresponding labels $Y = \{y_i\}_{i=1}^N$. In a typical regression problem for computer vision, $\mathbf{x}_i$ is an image or video, while $y_i \in \mathcal{R}^{\mathcal{Y}}$ takes a continuous value in the label space $\mathcal{Y}$. The target of regression is to recover $y_i$ by encoding the image to feature $\mathbf{z}_i = \varphi(\mathbf{x}_i)$ with encoder $\varphi$ and then mapping $\mathbf{z}_i$ to a predicted target $\hat{y}_i = f_\theta(\mathbf{z}_i)$ with a regression function $f(\cdot)$ parameterized by $\theta$. The encoder $\varphi$ and $\theta$ are learned by minimizing a regression loss function such as the mean-squared error $\mathcal{L}_{\mathrm{mse}} = \frac{1}{N} \sum_{i=1}^N (y_i - \hat{y}_i)^2$.

To formulate regression as a classification task with $K$ classes, the continuous target $Y$ can be converted to classes $Y^C = \{\mathbf{y}_i^c\}_{i=1}^N$ with some discretizing mapping function, where $\mathbf{y}_i^c \in [0, K-1]$ is a categorical target. The feature $\mathbf{z}_i$ is then mapped to the categorical target $\mathbf{y}_i^c = g_\omega(\mathbf{z}_i)$ with classifier $g_\omega(\cdot)$ parameterized by $\omega$. The encoder $\phi$ and $\omega$ are learned by minimizing the cross-entropy loss $\mathcal{L}_{CE} = -\frac{1}{N} \sum_{i=1}^N \log g_\omega(\mathbf{z}_i)$, where $(g_\omega(\mathbf{z}_i))_k = \frac{exp\omega_k^\mathsf{T} \mathbf{z}_i}{\sum_j exp\omega_k^\mathsf{T} \mathbf{z}_j}$ . Fig. 2(a) visualizes the symmetry of the two formulations.

The entropy of a random variable can be loosely defined as the amount of "information" associated with that random variable. One approach for estimating entropy $\mathcal{H}(Z)$ for a random variable $Z$ is the meanNN entropy estimator (Faivishevsky & Goldberger, 2008). It can accommodate higher-dimensional $Z$s, and is commonly used in high-dimensional space (Faivishevsky & Goldberger, 2010). The meanNN estimator relies on the distance between samples to approximate $p(Z)$. For a $D$-dimensional $Z$, it is defined as

$$\hat{\mathcal{H}}(Z) = \frac{D}{N(N-1)} \sum_{i \neq j} \log ||\mathbf{z}_i - \mathbf{z}_j||^2 + \mathrm{const}. \tag{1}$$

## 3.2 Feature Learning with a Cross-Entropy Loss (Classification)

Given that mutual information between target $Y^C$ and feature $Z$ is defined as $\mathcal{I}(Z; Y^C) = \mathcal{H}(Z) - \mathcal{H}(Z|Y^C)$, it follows that $\mathcal{I}(Z; Y^C)$ can be maximized by minimizing the second term $\mathcal{H}(Z|Y^C)$ and maximizing the first term $\mathcal{H}(Z)$. Boudiaf et al. (2020) showed that minimizing the cross-entropy loss accomplishes both by approximating the standard cross-entropy loss $\mathcal{L}_{CE}$ with a pairwise cross-entropy loss $\mathcal{L}_{PCE}$. $\mathcal{L}_{PCE}$ serves as a lower bound for $\mathcal{L}_{CE}$, and can be defined as

$$\mathcal{L}_{PCE} = \underbrace{-\frac{1}{2\lambda N^2} \sum_{i=1}^{N} \sum_{j \in [\mathbf{y}_j^c = \mathbf{y}_i^c]} \mathbf{z}_i^\mathsf{T} \mathbf{z}_j}_{\textit{Tightness} \propto \mathcal{H}(Z|Y^C)} + \underbrace{\frac{1}{N} \sum_{i=1}^{N} \log \sum_{k=1}^{K} \exp(\frac{1}{\lambda N} \sum_{j=1}^{N} p_{jk} \mathbf{z}_i^\mathsf{T} \mathbf{z}_j) - \frac{1}{2\lambda} \sum_{k=1}^{K} ||\mathbf{c}_k||}_{\textit{Diversity} \propto \mathcal{H}(Z)}, \quad (2)$$

where $\lambda \in \mathbb{R}$ is to make sure $\mathcal{L}_{CE}$ is a convex function with respect to $\omega$. Intuitively, $\mathcal{L}_{PCE}$ can be understood as being composed of both a pull and a push objective more familiar in contrastive learning. We interpret the pulling force as a tightness term. It encourages higher values for $\mathbf{z}_i^\mathsf{T} \mathbf{z}_j$ and closely aligns the feature vectors within a given class. This results in features clustered according to their class, *i.e.*, lower conditional entropy $\mathcal{H}(Z|Y^C)$. The pushing force from the second term encourages lower $\mathbf{z}_i^\mathsf{T} \mathbf{z}_j$ while forcing classes' centers $\mathbf{c}_k^s$ to be far from the origin. This results in diverse features that are spread apart, or higher marginal entropy $\mathcal{H}(Z)$. Note that the tightness term corresponds to the numerator of the Softmax function in $\mathcal{L}_{CE}$, while the diversity term corresponds to the denominator.

## 3.3 Feature Learning with an $\mathcal{L}_{\text{MSE}}$ Loss (Regression)

In this work, we find that minimizing $\mathcal{L}_{\text{mse}}$, as done in regression, is a proxy for minimizing $\mathcal{H}(Z|Y)$, without increasing $\mathcal{H}(Z)$. Minimizing $\mathcal{L}_{\text{mse}}$ does not increase the marginal entropy $\mathcal{H}(Z)$ and therefore limits feature diversity. The link between classification and regression is first established below in Lemma 1. We assume that we are dealing with a linear regressor, as is commonly used in deep neural networks.

**Lemma 1** *We are given dataset $\{\mathbf{x}_i, y_i\}_{i=1}^{N}$, where $\mathbf{x}_i$ is the input and $y_i \in \mathcal{Y}$ is the label, and linear regressor $f_\theta(\cdot)$ parameterized by $\theta$. Let $\mathbf{z}_i$ denote the corresponding feature. Assume that the label space $\mathcal{Y}$ is discretized into bins with maximum width $\eta$, and $c_i$ is the center of the bin to which $y_i$ belongs. Then for any $\epsilon > 0$, there exists $\eta > 0$ such that:*

$$|\mathcal{L}_{mse} - \frac{1}{N} \sum_{1}^{N} (\theta^\mathsf{T} \mathbf{z}_i - c_i)^2| \leq \frac{\eta}{2n} \sum_{1}^{n} |2\theta^\mathsf{T} \mathbf{z}_i - c_i - y_i| < \epsilon. \quad (3)$$

The detailed proof of Lemma 1 is provided in Appendix A.

The result of Lemma 1 says that the discretization error from replacing a regression target $y_i$ with $c_i$ can be made arbitrarily small if the bin width $\eta$ is sufficiently fine. As such, the $\mathcal{L}_{\text{mse}}$ can be directly approximated by the second term of Eq. 3 *i.e.*, $\mathcal{L}_{\text{mse}} \approx \frac{1}{N} \sum_{1}^{N} (\theta^\mathsf{T} \mathbf{z}_i - c_i)^2$. With this result, it can be proven that minimizing $\mathcal{L}_{\text{mse}}$ is a proxy for minimizing $\mathcal{H}(Z|Y)$.

**Theorem 1** *Let $\mathbf{z}_{c_i}$ denote the center of the features corresponding to bin center $c_i$, and $\phi_i$ be the angle between $\theta$ and $\mathbf{z}_i - \mathbf{z}_{c_i}$. Assume that $\theta$ is normalized, $(Z^c|Y) \sim \mathcal{N}(\mathbf{z}_{c_i}, I)$, where $Z^c$ is the distribution of $\mathbf{z}_{c_i}$ and that $\cos \phi_i$ is fixed. Then, minimizing $\mathcal{L}_{mse}$ can be seen as a proxy for minimizing $\mathcal{H}(Z|Y)$ without increasing $\mathcal{H}(Z)$.*

**Proof** Based on Lemma 1, we have

$$\mathcal{L}_{\text{mse}} = \frac{1}{N} \sum_{1}^{N} (\theta^\mathsf{T} (\mathbf{z}_i - \mathbf{z}_{c_i}))^2 = \frac{1}{N} \sum_{1}^{N} (||\theta|| ||\mathbf{z}_i - \mathbf{z}_{c_i}|| \cos \phi_i)^2$$

$$= \frac{1}{N} \sum_{1}^{N} ||\theta||^2 ||\mathbf{z}_i - \mathbf{z}_{c_i}||^2 \cos^2 \phi_i \propto \frac{1}{N} \sum_{1}^{N} ||\mathbf{z}_i - \mathbf{z}_{c_i}||^2. \quad (4)$$

Note, $\mathbf{z}_{c_i}$ exist unless $\theta = 0$ and $c_i \neq 0$. Since it is assumed that $Z^c|Y \sim \mathcal{N}(\mathbf{z}_{c_i}, I)$, the term $\frac{1}{N} \sum_1^N ||\mathbf{z}_i - \mathbf{z}_{c_i}||^2$ can be interpreted as a conditional cross entropy between $Z$ and $Z^c$, as it satisfies

$$\mathcal{H}(Z; Z^c|Y) = -\mathbb{E}_{\mathbf{z} \sim Z|Y}[\log p_{Z^c|Y}(\mathbf{z})] \overset{mc}{\approx} \frac{-1}{N} \sum_{i=1}^N \log(e^{\frac{-1}{2}||\mathbf{z}_i - \mathbf{z}_{c_i}||^2}) + \text{const}$$

$$\overset{c}{=} \frac{1}{N} \sum_{i=1}^N ||\mathbf{z}_i - \mathbf{z}_{c_i}||^2, \tag{5}$$

where $\overset{c}{=}$ denotes equal to, up to a multiplicative and an additive constant. The $\overset{mc}{\approx}$ denotes Monte Carlo sampling from the $Z|Y$ distribution, allowing us to replace the expectation by the mean of the samples. Subsequently, we can show that

$$\mathcal{L}_{\text{mse}} \propto \frac{1}{N} \sum_1^N ||\mathbf{z}_i - \mathbf{z}_{c_i}||^2 \overset{c}{=} \mathcal{H}(Z; Z^c|Y) = \mathcal{H}(Z|Y) + \mathcal{D}_{KL}(Z||Z^c|Y). \tag{6}$$

The result in Eq. 6 shows that $\frac{1}{N} \sum_1^N ||\mathbf{z}_i - \mathbf{z}_{c_i}||^2$ is an upper bound of the tightness term in mutual information. If $(Z|Y) \sim \mathcal{N}(\mathbf{z}_{c_i}, I)$, then $\mathcal{D}_{KL}(Z||Z^c|Y)$ is equal to 0 and the bound is tight i.e., $\frac{1}{N} \sum_1^N ||\mathbf{z}_i - \mathbf{z}_{c_i}||^2 \geq \mathcal{H}(Z|Y)$. Hence, minimizing $\mathcal{L}_{\text{mse}}$ is a proxy for minimizing $\mathcal{H}(Z|Y)$.

Apart from $\mathcal{H}(Z|Y)$, the relation in Eq. 6 also contains the KL divergence between the two conditional distributions $P(Z|Y)$ and $P(Z^c|Y)$, where $Z_i^c$ are feature centers of Z. Minimizing this divergence will either force $Z$ closer to the centers $Z^c$, or move the centers $Z^c$ around. By definition, however, the cluster centers $Z_i^c$ cannot expand beyond $Z$'s coverage, so features Z must shrink to minimize the divergence. As such, the entropy $\mathcal{H}(Z)$ will not be increased by this term. $\square$

Based on Eq. 2 and Theorem 1, we draw the conclusion that regression, with an MSE loss, overlooks the marginal entropy $\mathcal{H}(Z)$ and results in a less diverse feature space than classification with a cross-entropy loss.

It is worth mentioning that the Gaussian distribution assumption, *i.e.*, $Z^c|Y \sim \mathcal{N}(\mathbf{z}_{c_i}, I)$, is standard in the literature when analyzing features (Yang et al., 2021a; Salakhutdinov et al., 2012) and entropy (Misra et al., 2005), and $\cos \phi_i$ is a constant value at each iteration.

## 4 ORDINAL ENTROPY

Our theoretical analysis in Sec. 3 shows that learning with only the MSE loss does not increase the marginal entropy $\mathcal{H}(Z)$ and results in lower feature diversity. To remedy this situation, we propose a novel regularizer to encourage a higher entropy feature space.

Using the distance-based entropy estimate from Eq. 1, one can then minimize the the negative distances between feature centers $\mathbf{z}_{c_i}$ to maximize the entropy of the feature space. $\mathbf{z}_{c_i}$ are calculated by taking a mean over all the features $\mathbf{z}$ which project to the same $y_i$. Note that as feature spaces are unbounded, the features $\mathbf{z}$ must first be normalized, and below, we assume all the features $\mathbf{z}$ are already normalized with an L2 norm:

$$\mathcal{L}'_d = -\frac{1}{M(M-1)} \sum_{i=1}^M \sum_{i \neq j} ||\mathbf{z}_{c_i} - \mathbf{z}_{c_j}||_2 \propto -H(Z), \tag{7}$$

where $M$ is the number of feature centers in a batch of samples or a sampled subset sampled from a batch. We consider each feature as a feature center when the continuous labels of the dataset are precise enough.

While the regularizer $\mathcal{L}'_d$ indeed spreads features to a larger extent, it also breaks ordinality in the feature space (see Fig. 3(b)). As such, we opt to weight the feature norms in $\mathcal{L}'_d$ with $w_{ij}$, where $w_{ij}$ are the distances in the label space $\mathcal{Y}$:

$$\mathcal{L}_d = -\frac{1}{M(M-1)} \sum_{i=1}^M \sum_{i \neq j} w_{ij} ||\mathbf{z}_{c_i} - \mathbf{z}_{c_j}||_2, \qquad \text{where } w_{ij} = ||y_i - y_j||_2 \tag{8}$$

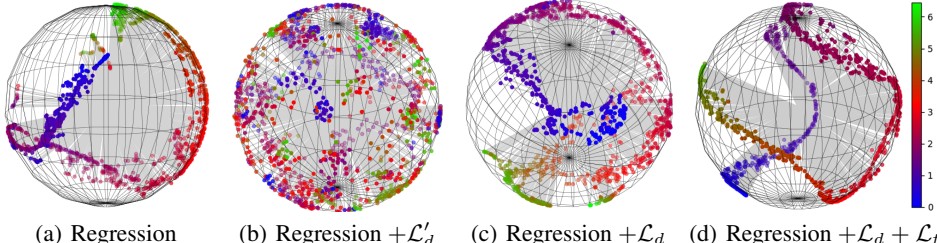

| (a) Regression | (b) Regression $+\mathcal{L}'_d$ | (c) Regression $+\mathcal{L}_d$ | (d) Regression $+\mathcal{L}_d + \mathcal{L}_t$ |

Figure 3: t-SNE visualization of features from the depth estimation task. (b) Simply spreading the features ($\mathcal{L}'_d$) leads to a higher entropy feature space, while the ordinal relationship is lost. (c) By further exploiting the ordinal relationship in the label space ($\mathcal{L}_d$), the features are spread and the ordinal relationship is also preserved. (d) Adding the tightness term ($\mathcal{L}_t$) further encourages features close to its centers.

As shown in Figure 3(c), $\mathcal{L}_d$ spreads the feature while also preserve preserving ordinality. Note that $\mathcal{L}'_d$ is a special case of $\mathcal{L}_d$ when $w_{ij}$ are all equal.

To further minimize the conditional entropy $\mathcal{H}(Z|Y)$, we introduce an additional tightness term that directly considers the distance between each feature $\mathbf{z}_i$ with its centers $\mathbf{z}_{c_i}$ in the feature space:

$$\mathcal{L}_t = \frac{1}{N_b} \sum_{i=1}^{N_b} ||\mathbf{z}_i - \mathbf{z}_{c_i}||_2, \tag{9}$$

where $N_b$ is the batch size. Adding this tightness term further encourages features close to its centers (compare Fig. 3(c) with Fig. 3(d)). Compared with features from standard regression (Fig. 3(a)), the features in Fig. 3(d) are more spread, *i.e.*, the lines formed by the features are longer.

We define the ordinal entropy regularizer as $\mathcal{L}_{oe} = \mathcal{L}_d + \mathcal{L}_t$, with a diversity term $\mathcal{L}_d$ and a tightness term $\mathcal{L}_t$. $\mathcal{L}_{oe}$ achieves similar effect as classification in that it spreads $\mathbf{z}_{c_i}$ while tightening features $z_i$ corresponding to $\mathbf{z}_{c_i}$. Note, if the continuous labels of the dataset are precise enough and each feature is its own center, then our ordinal entropy regularizer will only contain the diversity term, *i.e.*, $\mathcal{L}_{oe} = \mathcal{L}_d$. We show our regression with ordinal entropy (red dotted arrow) in Fig. 2(a).

The final loss function $\mathcal{L}_{total}$ is defined as:

$$\mathcal{L}_{total} = \mathcal{L}_m + \lambda_d \mathcal{L}_d + \lambda_t \mathcal{L}_t, \tag{10}$$

where $\mathcal{L}_m$ is the task-specific regression loss and $\lambda_d$ and $\lambda_e$ are trade-off parameters.

## 5 EXPERIMENTS

### 5.1 DATASETS, METRICS & BASELINE ARCHITECTURES

We conduct experiments on four tasks: operator learning, a synthetic dataset and three real-world regression settings of depth estimation, crowd counting, age estimation.

For **operator learning**, we follow the task from DeepONet (Lu et al., 2021) and use a two-layer fully connected neural network with 100 hidden units. See Sec. 5.2 for details on data preparation.

For **depth estimation**, NYU-Depth-v2 (Silberman et al., 2012) provides indoor images with the corresponding depth maps at a pixel resolution $640 \times 480$. We follow (Lee et al., 2019) and use ResNet50 (He et al., 2016) as our baseline architecture unless otherwise indicated. We use the train/test split given used by previous works (Bhat et al., 2021; Yuan et al., 2022), and evaluate with the standard metrics of threshold accuracy $\delta_1$, average relative error (REL), root mean squared error (RMS) and average $\log_{10}$ error.

For **crowd counting**, we evaluate on SHA and SHB of the ShanghaiTech crowd counting dataset (SHTech) (Zhang et al., 2015). Like previous works, we adopt density maps as labels and evaluate with mean absolute error (MAE) and mean squared error (MSE). We follow Li et al. (2018) and use CSRNet with ResNet50 as the regression baseline architecture.

Table 1: Ablation studies on linear and nonlinear operators learning with synthetic data and depth estimation on NYU-Depth-v2. For operator learning, we report results as mean $\pm$ standard variance over 10 runs. **Bold** numbers indicate the best performance.

| Method | Linear | Nonlinear | NYU-Depth-v2 | | | |
|---|---|---|---|---|---|---|
| | $\mathcal{L}_{\mathrm{mse}}(\times 10^{-3})\downarrow$ | $\mathcal{L}_{\mathrm{mse}}(\times 10^{-2})\downarrow$ | $\delta_1\uparrow$ | REL $\downarrow$ | RMS $\downarrow$ | $\log_{10}\downarrow$ |
| Baseline | $3.0\pm0.93$ | $2.5\pm2.0$ | 0.793 | 0.148 | 0.502 | 0.064 |
| Baseline + $\mathcal{L}'_d$ | $2.2\pm0.42$ | $0.8\pm0.3$ | 0.795 | 0.147 | 0.505 | 0.063 |
| Baseline + $\mathcal{L}_d$ | $\mathbf{1.6\pm0.34}$ | $0.5\pm0.3$ | 0.808 | 0.144 | 0.483 | 0.061 |
| Baseline + $\mathcal{L}_d + \mathcal{L}_t$ | - | - | **0.811** | **0.143** | **0.478** | **0.060** |
| w/ cosine distance | $1.7\pm0.33$ | $0.6\pm0.5$ | 0.806 | 0.147 | 0.488 | 0.061 |
| w/o normalization | $5.3\pm1.10$ | $15.5\pm0.4$ | 0.790 | 0.153 | 0.510 | 0.064 |
| $w_{ij}=\|y_i-y_j\|_2^2$ | $2.0\pm0.60$ | $\mathbf{0.4\pm0.2}$ | 0.800 | 0.149 | 0.498 | 0.063 |
| $w_{ij}=\sqrt{\|y_i-y_j\|_2}$ | $1.8\pm0.50$ | $0.5\pm0.2$ | 0.799 | 0.147 | 0.496 | 0.062 |

For **age estimation**, we use AgeDB-DIR (Yang et al., 2021b) and also implement their regression baseline model, which uses ResNet-50 as a backbone. Following Liu et al. (2019b), we report results on three disjoint subsets (*i.e.*, ,Many, Med. and Few), and also overall performance (*i.e.*, ALL). We evaluate with MAE and geometric mean (GM).

**Other Implementation Details:** We follow the settings of previous works DeepONet (Lu et al., 2021) for operator learning, Adabins (Bhat et al., 2021) for depth estimation, CSRNet (Li et al., 2018) for crowd counting, and Yang et al. (2021b) for age estimation. See Appendix D for details.

$\lambda_d$ and $\lambda_t$ are set empirically based on the scale of the task loss $\lambda_m$. We use the trade-off parameters $\lambda_d, \lambda_t$ the same value of $0.001, 1, 10, 1$, for operator learning, depth estimation, crowd counting and age estimation, respectively.

## 5.2 LEARNING LINEAR AND NONLINEAR OPERATORS

We first verify our method on the synthetic task of operator learning. In this task, an (unknown) operator maps input functions into output functions and the objective is to regress the output value. We follow (Lu et al., 2021) and generate data for both a linear and non-linear operator. For the linear operator, we aim to learn the integral operation $G$:

$$G : u(x) \mapsto s(x) = \int_0^x u(\tau)d\tau, x \in [0,1], \tag{11}$$

where $u$ is the input function, and $s$ is the target function. The data is generated with a mean-zero Gaussian random field function space: $u \sim \mathcal{G}(0, k_l(x_1, x_2))$, where the covariance kernel $k_l(x_1, x_2) = exp(-\|x_1 - x_2\|^2/2l^2)$ is the radial-basis function kernel with a length-scale parameter $l = 0.2$. The function $u$ is represented by the function values of $m = 100$ fixed locations $\{x_1, x_2, \cdots, x_m\}$. The data is generated as $([u, y], G(u)(y))$, where $y$ is sampled from the domain of $G(u)$. We randomly sample 1k data as the training set and test on the testing set with 100k samples.

For the nonlinear operator, we aim to learn the following stochastic partial differential equation, which maps $b(x; \omega)$ of different correlation lengths $l \in [1, 2]$ to a solution $u(x; \omega)$:

$$\mathrm{div}(e^{b(x;\omega)}\nabla u(x;\omega)) = f(x), \tag{12}$$

where $x \in (0, 1)$, $\omega$ from the random space with Dirichlet boundary conditions $u(0) = u(1) = 0$, and $f(x) = 10$. The randomness comes from the diffusion coefficient $e^{b(x;\omega)}$. The function $b(x; \omega) \sim \mathcal{GP}(b_0(x), \mathrm{cov}(x1, x2))$ is modelled as a Gaussian random process $\mathcal{GP}$, with mean $b_0(x) = 0$ and $\mathrm{cov}(x1, x2) = \sigma^2 \exp(-\|x_1 - x_2\|^2/2l^2)$. We randomly sample 1k training samples and 10k test samples.

For operator learning, we set $\mathcal{L}_{\mathrm{mse}}$ as the task-specific baseline loss for both the linear and non-linear operator. Table 1 shows that even without ordinal information, adding the diversity term *i.e.*, $\mathcal{L}'_d$ to $\mathcal{L}_{\mathrm{mse}}$ already improves performance. The best gains, however, are achieved by incorporating the weighting with $\mathcal{L}_d$, which decreases $\mathcal{L}_{\mathrm{mse}}$ by 46.7% for the linear operator and up to 80% for the more challenging non-linear operator. The corresponding standard variances are also reduced significantly.

Table 2: Quantitative comparison of depth estimation results with NYU-Depth-v2. **Bold** numbers indicate the best performance.

| Method | $\delta_1 \uparrow$ | REL $\downarrow$ | RMS $\downarrow$ | $\log_{10} \downarrow$ |
|---|---|---|---|---|
| Laina et al. (Laina et al., 2016) | 0.811 | 0.127 | 0.573 | 0.055 |
| DORN (Fu et al., 2018) | 0.828 | 0.115 | 0.509 | 0.051 |
| BTS (Lee et al., 2019) | 0.885 | 0.110 | 0.392 | 0.047 |
| Adabins (Bhat et al., 2021) | 0.903 | 0.103 | 0.364 | 0.044 |
| NeW-CRFs (Yuan et al., 2022) | 0.922 | 0.095 | 0.334 | 0.041 |
| NeW-CRFs + $\mathcal{L}_d$ + $\mathcal{L}_t$ | **0.932** | **0.089** | **0.321** | **0.039** |
| Baseline (ResNet-50) | 0.793 | 0.148 | 0.502 | 0.064 |
| Baseline (ResNet-50) + $\mathcal{L}_d$ | 0.808 | 0.144 | 0.483 | 0.061 |
| Baseline (ResNet-50) + $\mathcal{L}_d$ + $\mathcal{L}_t$ | 0.811 | 0.143 | 0.478 | 0.060 |

Table 3: Results on SHTech. **Bold** numbers indicate the best performance.

| Method | MAE $\downarrow$ | | MSE $\downarrow$ | |
|---|---|---|---|---|
| | SHA | SHB | SHA | SHB |
| Regression Baseline (Li et al., 2018) | 68.2 | 10.6 | 115.0 | 16.0 |
| +$\mathcal{L}_d$ | 66.9 | **9.1** | 107.5 | 14.7 |
| +$\mathcal{L}_d$ + $\mathcal{L}_t$ | **65.6** | **9.1** | **105.0** | **14.5** |

Note that we do not verify $\mathcal{L}_t$ on operator learning due to the high data precision on synthetic datasets, it is difficult to sample points belonging to the same $\mathbf{z}_{c_i}$. Adding $\mathcal{L}_t$, however, is beneficial for the three real-world tasks (see Sec. 5.3).

## 5.3 REAL-WORLD TASKS: DEPTH ESTIMATION, CROWD COUNTING & AGE ESTIMATION

**Depth Estimation:** Table 2 shows that adding the ordinal entropy terms boosts the performance of the regression baseline and the state-of-the-art regression method NeW-CRFs (Yuan et al., 2022). NeW-CRFs with ordinal entropy achieves the highest values for all metrics, decreasing $\delta_1$ and REL errors by 12.8% and 6.3%, respectively. Moreover, higher improvement can be observed when adding the ordinal entropy into a simpler baseline, *i.e.*, ResNet-50.

**Crowd Counting:** Table 3 shows that adding $\mathcal{L}_d$ and $\mathcal{L}_t$ each contribute to improving the baseline. Adding both terms has the largest impact and for SHB, the improvement is up to 14.2% on MAE and 9.4% on MSE.

**Age Estimation:** Table 4 shows that with $\mathcal{L}_d$ we achieve a significant 0.13 and 0.29 overall improvement (*i.e.*, ALL) on MAE and GM, respectively. Applying $\mathcal{L}_t$ achieves a further overall improvement over $\mathcal{L}_d$ only, including 0.14 on MAE and 0.04 on GM.

## 5.4 ABLATION STUDIES

Ablation results on both operator learning and depth estimation are shown in Table 1 and Figure 4.

**Ordinal Relationships:** Table 1 shows that using the unweighted diversity term 'Baseline+$\mathcal{L}_d'$', which ignores ordinal relationships, is worse than the weighted version 'Baseline+$\mathcal{L}_d$' for both operator learning and depth estimation.

**Feature Normalization:** As expected, normalization is important, as performance will decrease (compare 'w/o normalization' to 'Baseline+$\mathcal{L}_d$' in Table 1) for both operator learning and depth estimation. Most interestingly, normalization also helps to lower variance for operator learning.

**Feature Distance** $||\mathbf{z}_{c_i} - \mathbf{z}_{c_j}||$: We replace the original feature distance L2 with cosine distance (see 'w/ cosine distance') and the cosine distance is slightly worse than L2 for all the cases.

**Weighting Function** $w_{ij}$: The weight as defined in Eq. 8 is based on an L2 distance. Table 1 shows that L2 is best for linear operator learning and depth estimation but is slightly worse than $w_{ij} = ||y_i - y_j||_2^2$ for nonlinear operator learning.

Table 4: Results on AgeDB-DIR. **Bold** numbers indicate the best performance.

| Method | MAE ↓ | | | | GM ↓ | | | |
|---|---|---|---|---|---|---|---|---|
| | ALL | Many | Med. | Few | ALL | Many | Med. | Few |
| Baseline (Yang et al., 2021b) | 7.77 | **6.62** | 9.55 | 13.67 | 5.05 | 4.23 | 7.01 | 10.75 |
| $+\mathcal{L}_d$ | 7.60 | 6.79 | 8.55 | 12.70 | 4.76 | 4.15 | 5.95 | **9.60** |
| $+\mathcal{L}_d + \mathcal{L}_t$ | **7.46** | 6.73 | **8.18** | **12.38** | **4.72** | 4.21 | **5.36** | 9.70 |

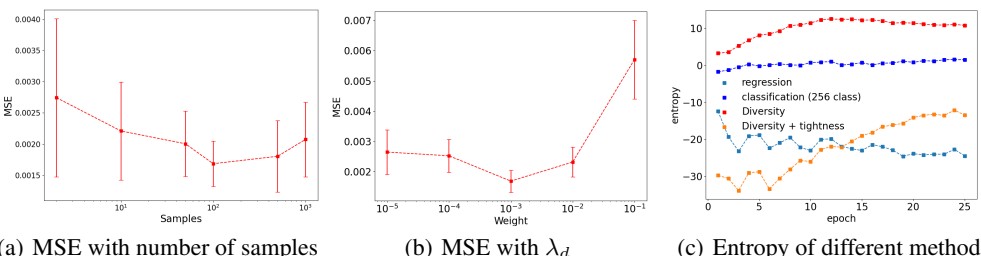

(a) MSE with number of samples     (b) MSE with $\lambda_d$     (c) Entropy of different methods

Figure 4: Based on the linear operator learning and depth estimation, we show (a) the effect of the number of samples on MSE, (b) the performance analysis with different $\lambda_d$ and (c) the entropy curves of different methods during testing. The results for the nonlinear operator learning are given in Appendix C.

**Sample Size (M)** In practice, we estimate the entropy from a limited number of regressed samples and this is determined by the batch size. For certain tasks, this may be sufficiently large, *e.g.*, depth estimation (number of pixels per image × batch size) or very small, *e.g.*, age estimation (batch size). We investigate the influence of $M$ from Eq. 8 on linear operator learning (see Fig.4(b)). In the most extreme case, when $M = 2$, the performance is slightly better than the baseline model ($2.7 \times 10^{-3}$ vs. $3.0 \times 10^{-3}$), suggesting that our ordinal regularizer terms are effective even with 2 samples. As $M$ increases, the MSE and its variance steadily decrease as the estimated entropy likely becomes more accurate. However, at a certain point, MSE and variance start to increase again. This behavior is not surprising; with too many samples under consideration, it likely becomes too difficult to increase the distance between a pair of points without decreasing the distance to other points, *i.e.*, there is not sufficient room to maneuver. The results for the nonlinear operator learning are given in Appendix C.

**Hyperparameter $\lambda_d$ and $\lambda_t$:** Fig. 4(b) plots the MSE for linear operator learning versus the trade-off hyper-parameter $\lambda_d$ applied to the diversity term $\mathcal{L}_d$. Performance remains relatively stable up to $10^{-2}$, after which this term likely overtakes the original learning objective $\mathcal{L}_{\text{mse}}$ and causes MSE to decrease. The results for the nonlinear operator and analysis on $\lambda_t$ are given in Appendix C and E.

**Marginal Entropy $\mathcal{H}(Z)$:** We show the marginal entropy of the testing set from different methods during training (see Fig. 4(c)). We can see that the marginal entropy of classification is always larger than that of regression, which has a downward trend. Regression with only diversity achieves the largest marginal entropy, which verifies the effectiveness of our diversity term. With both diversity and tightness terms, as training goes, its marginal entropy continues to increase and larger than that of regression after the $13^{th}$ epoch. More experiment results can be found in Appendix F.

## 6 CONCLUSION

In this paper, we dive deeper into the trend of solving regression-type problems as classification tasks by comparing the difference between regression and classification from a mutual information perspective. We conduct a theoretical analysis and show that regression with an MSE loss lags in its ability to learn high-entropy feature representations. Based on the findings, we propose an ordinal entropy regularizer for regression, which not only keeps an ordinal relationship in feature space like regression, but also learns a high-entropy feature representation like classification. Experiments on different regression tasks demonstrate that our entropy regularizer can serve as a plug-in component for regression-based methods to further improve the performance.

**Acknowledgement**. This research / project is supported by the Ministry of Education, Singapore, under its MOE Academic Research Fund Tier 2 (STEM RIE2025 MOE-T2EP20220-0015).

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

**Appendix**

## A  PROOF OF LEMMA 1

**Proof**

$$\mathcal{L}_{\text{mse}} = \frac{1}{n}\sum_1^n (\theta^\mathsf{T}\mathbf{z}_i - y_i)^2$$

$$= \frac{1}{n}\sum_1^n ((\theta^\mathsf{T}\mathbf{z}_i - c_i) + (c_i - y_i))^2$$

$$= \frac{1}{n}\sum_1^n (\theta^\mathsf{T}\mathbf{z}_i - c_i)^2 + \frac{1}{n}\sum_1^n ((c_i - y_i)(2\theta^\mathsf{T}\mathbf{z}_i - c_i - y_i))$$

Since $|c_i - y_i| \leq \frac{\eta}{2}$, we have:

$$\left|\mathcal{L}_{\text{mse}} - \frac{1}{n}\sum_1^n (\theta^\mathsf{T}\mathbf{z}_i - c_i)^2\right| \leq \frac{\eta}{2n}\sum_1^n |2\theta^\mathsf{T}\mathbf{z}_i - c_i - y_i|$$

For $\eta < \frac{2n\epsilon}{\sum_1^n |2\theta^\mathsf{T}\mathbf{z}_i - c_i - y_i|}$:

$$\left|\mathcal{L}_{\text{mse}} - \frac{1}{n}\sum_1^n (\theta^\mathsf{T}\mathbf{z}_i - c_i)^2\right| < \epsilon$$

$\square$

## B  VISUALIZATION

**Experimental setting** We train the regression and classification models on the NYU-Depth-v2 dataset for depth estimation. We modify the last layer of a ResNet-50 model to a convolution operation with kernel size $1 \times 1$, and train the modified model with $\mathcal{L}_{\text{mse}}$ as our regression model. For the classification models, we modify the last layer of two ResNet-50 models to output $N_c$ channels, where $N_c$ is the number of classes, and train the modified models with cross-entropy. The classes are defined by uniformly discrete ground-truth depths into $N_c$ bins. The entropy of feature space is estimated using Eq 1 on pixel-wise features over the training and test set of NYU-Depth-v2. After training, we visualize the pixel-wise features of an image from the test set using t-distributed stochastic neighbor embedding (t-SNE), and features are colored based on their predicted depth.

The visualization results are shown in Figure A. We exploit three entropy estimators to estimate the entropy of feature space $\mathcal{H}(Z)$. Entropy in the first row of Figure A is estimated with the meanNN entropy estimator Eq. 1. Entropy in the second row is also estimated with the meanNN entropy estimator, where the input is the features normalized with the L2 norm. Entropy in the third row is estimated with the diversity part of our ordinal entropy Eq. 8.

We make several interesting observations from the visualization results: (1) On both training and testing, compared with classification, regression always has a lower estimated entropy based on all three entropy estimators; (2) benefiting from the diversity part of our ordinal entropy $\mathcal{L}_d$, regression produces a higher entropy space than the classification model; (3) $\mathcal{L}_d$ with the tightness term $\mathcal{L}_t$ can result in feature space with a similarly estimated entropy compared with classification.

## C  RESULTS ON THE NONLINEAR OPERATOR LEARNING PROBLEM

We analyze the effect of the number of samples on MSE loss and the performance with $\lambda_d$ on the nonlinear operator learning task. The results are shown in Figure B. The results corroborate the conclusions derived from the linear task.

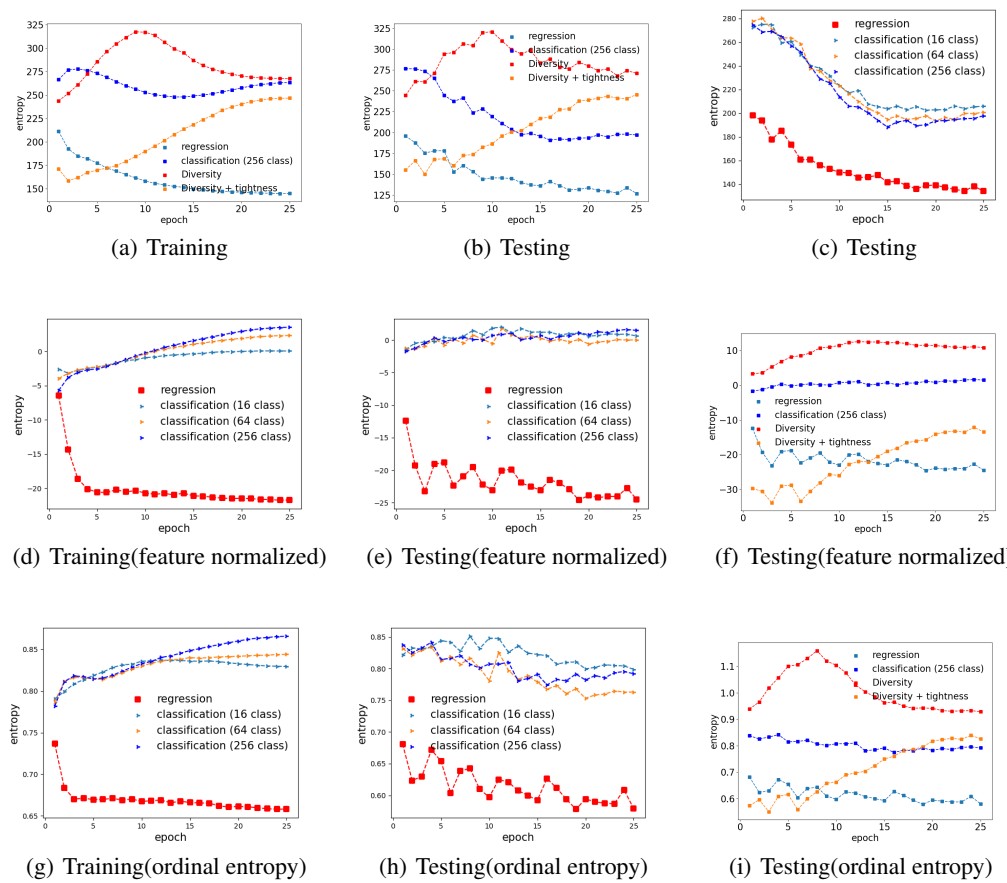

Figure A: Visualization results with different entropy estimators on training and testing set.

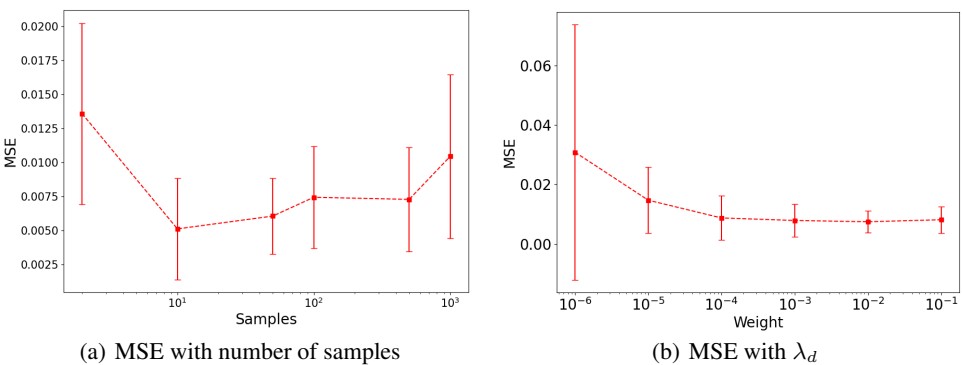

Figure B: Based on the nonlinear operator learning problem, we show (a) the effect of the number of samples on MSE loss, (b) the performance analysis with different $\lambda_d$.

# D  EVALUATION METRICS

Here we introduce the definition of the evaluation metrics for depth estimation, crowd counting, and age estimation.

**Depth Estimation.** We denote the predicted depth at position $p$ as $y_p$ and the corresponding ground truth depth as $y_p'$, the total number of pixels is $n$. The metrics are: 1) threshold accuracy $\delta_1 \triangleq \%$ of $y_p$, s.t. $\max(\frac{y_p}{y_p'}, \frac{y_p'}{y_p}) < t_1$, where $t_1 = 1.25$; 2) average relative error (REL): $\frac{1}{n} \sum_p \frac{|y_p - y_p'|}{y_p}$; 3) root mean squared error (RMS): $\sqrt{\frac{1}{n} \sum_p (y_p - y_p')^2}$; 4) average ($\log_{10}$ error): $\frac{1}{n} \sum_p |\log_{10}(y_p) - \log_{10}(y_p')|$.

**Crowd Counting.** Given $N$ images for testing, $y_i, y_i'$ are the estimated count and the ground truth for the $i$-th image, respectively. We exploit two widely used metrics as measurements: 1) Mean Absolute Error (MAE)= $\frac{1}{N} \sum_{i=1}^{N} |y_i - y_i'|$, and 2) Mean Squared Error (MSE) = $\sqrt{\frac{1}{N} \sum_{i=1}^{N} |y_i - y_i'|^2}$.

**Age Estimation.** Given $N$ images for testing, $y_i$ and $y_i'$ are the $i$-th prediction and ground-truth, respectively. The evaluation metrics include 1)MAE: $\frac{1}{N} \sum_{i=1}^{N} |y_i - y_i'|$, and 2)Geometric Mean (GM): $(\prod_{i=1}^{N} |y_i - y_i'|)^{\frac{1}{N}}$.

## E    EFFECT OF $\lambda_t$

We analyze the effect of $\lambda_t$ with an ablation study on the age estimation with Age-DB-DIR. Table 5 shows that the final performance is not sensitive to the change of $\lambda_t$, and $\mathcal{L}_t$ is effective even with a small $\lambda_t$, *i.e.*, 0.1.

Table 5: Results on AgeDB-DIR. **Bold** numbers indicate the best performance.

| $\lambda_t$ | MAE ↓ | | | | GM ↓ | | | |
|---|---|---|---|---|---|---|---|---|
| | ALL | Many | Med. | Few | ALL | Many | Med. | Few |
| 0 | 7.60 | 6.79 | 8.55 | 12.70 | 4.76 | 4.15 | 5.95 | **9.60** |
| 0.1 | 7.50 | **6.53** | 8.64 | 13.50 | **4.71** | **4.04** | 5.92 | 10.62 |
| 1 | **7.46** | 6.73 | **8.18** | **12.38** | 4.72 | 4.21 | **5.36** | 9.70 |
| 10 | 7.49 | 6.70 | 8.27 | 12.86 | 4.79 | 4.23 | 5.61 | 10.21 |

## F    INFLUENCE OF THE SAMPLE SIZE (M)

Efficiency-wise, the computing complexity of the regularizer is quadratic with respect to $M$. The synthetic experiments on operator learning (Table 6) use a 2-layer MLP, so the regularizer adds significant computing time when M gets large. However, the real-world experiments on depth estimation (Table 7) use a ResNet-50 backbone, and the added time and memory are negligible (27% and 0.3%, respectively), even with $M = 3536$. We exploit $M = 3536$ in our depth estimation experiments, where 3536 is a 16x subsampling of the total number of pixels in an image. Note that these increases are only during training and do not add computing demands for inference. In addition, the added time and memory with $L_t$ are also negligible (0.08% and 0% , respectively), even with M=3536.

Table 6: Quantitative comparison of the time consumption and memory consumption on linear operator learning with synthetic data.

| M | Training time (s) | Memory taken (MB) |
|---|---|---|
| 0 | 155 | 2163 |
| 2 | 291 | 2163 |
| 10 | 296 | 2163 |
| 100 | 327 | 2163 |
| 1000 | 1033 | 2205 |

Table 7: Quantitative comparison of the time consumption and memory consumption on depth estimation with NYU-v2. The training time is one epoch training time.

| M | Regularizer | Training time (s) | Memory taken (MB) |
|---|---|---|---|
| 0 | no regularizer | 1836 | 12363 |
| 100 | $L_d + L_t$ | 1877 | 12379 |
| 1000 | $L_d + L_t$ | 1999 | 12395 |
| 3526 | $L_d$ | 2342 | 12405 |
| 3526 | $L_d + L_t$ | 2344 | 12405 |

