# OpenReview forum: "Improving Deep Regression with Ordinal Entropy"
_ICLR.cc/2023/Conference — ICLR 2023 poster_

### Official Review · Reviewer_eTrs · 2022-10-23

**Confidence:** 4
**Correctness:** 4
**Technical Novelty And Significance:** 3
**Empirical Novelty And Significance:** 3
**Recommendation:** 8

**Clarity, Quality, Novelty And Reproducibility:**

Clarity:

[+] The presentation of the idea is clear and easy to follow. The experimental demonstrations are also convincing and solid. However, the authors should proofread the manuscript and pay more attention to the typos.

Quality:

[+] The theoretical and empirical demonstrations are both clear and technically sound.

Originality:

[+] To the best of my knowledge, I think this work is novel and can be a contribution to the ML community.

Reproducibility:

It would greatly improve the contribution if the authors can release the code.

**Strength And Weaknesses:**

Strength:

(1) I like the analysis of regression and its reformulating classification through the lens of mutual information and entropy. It does theoretically explain the difference between regression penalized by MSE and regression penalized by cross-entropy. Besides the empirical improvement, I do value this theoretical finding.

(2) The idea is well-presented and easy to follow. The experimental demonstration is also convincing and extensive.

Weakness:

(1) The authors should consider conducting an ablation study on $\lambda_d$ and $\lambda_t$. Is the final performance sensitive to these two hyperparameters?

(2) Are Eq. (8) and (9) defined on the mini-batch or the entire dataset? How does adding tightness and diversity terms increase the overall training cost?

(3) Missing related work [1]. The idea of preserving ordinality by using the label distances is interesting and technically sound. A similar idea in the case of preserving ordinality in regression is also used in [1]. The authors should discuss their difference.

(4) Minor point: I encourage the authors to further proofread the manuscript. I can find some of the typos as follows:
- Section 3.1 P2 L3 ‘The simplest mapping The feature encoding’
- Sec. 3.2 P2 L2 ‘being a being composed’
- Introduction Paragraph 4 ‘and regress’ to ‘and regression’
- Section 4 P3 L2 ‘is precision enough’



[1] Gong et al., RankSim: Ranking Similarity Regularization for Deep Imbalanced Regression, ICML 2022

**Summary Of The Paper:**

The authors analyze the observation that reformulating regression as classification often yields better performance by better learning high-entropy feature representations. To retain the benefits of both high entropy and ordinality of feature learning, the authors propose an ordinal entropy loss to encourage higher-entropy feature spaces while maintaining ordinal relationships for regression tasks. Extensive experiments demonstrate the empirical improvement of the method.

**Summary Of The Review:**

The authors propose an ordinal entropy loss with the original regression task loss to improve the regression performance, based on the theoretical analysis that reformulating regression as a classification problem can often improve the performance. The method is also experimentally demonstrated on multiple different regression tasks. Therefore, I recommend acceptance.

---

> ### Author Response · Authors · 2022-11-19
> **Thanks and response to concerns**
>
> Thank you for the careful review and constructive suggestions.
>
> For the specific concerns:
>
> ---
>
> **Q1: Ablation study on $\lambda_t$ and $\lambda_d$.**
>
> **A1:** Thanks for pointing this out.
>
> $\lambda_d$:
>
> Figure 4(b) in our paper shows the impact of $\lambda_d$.  Performance steadily improves with larger $\lambda_d$, until it is too large and the regularization term dominates over the main MSE loss.
>
> $\lambda_t$:
>
> We add an ablation study for age estimation on Age-DB-DIR (Table 1) and observe a similar trend as $\lambda_d$.  Performance steadily improves, until the regularization term starts to overpower the main MSE loss at which point performance drops.
>
> Table 1.
>
> |             $\lambda_t$              | MAE (ALL) | GM (ALL) |
> |:----------------------------------:|:--------:|:-------:|
> |                 0                  |   7.60   |  4.76   |
> |                0.1                 |   7.50   |  4.71   |
> |                 1                  |   7.46   |  4.72   |
> |                 10                 |   7.49   |  4.79   |
>
> ---
>
> **Q2: Is the proposed regularizer defined on the mini-batch or the entire dataset? How does adding $L_d$ and $L_t$ increase the overall training cost?**
>
> **A2:** Thanks for pointing this out. Please see the general response G1.
>
> ---
>
> **Q3: Related works on ordinal regression:**
>
> **A3:** Thanks for pointing this out. Please see the general response G3.

---

### Official Review · Reviewer_ejHX · 2022-10-25

**Confidence:** 4
**Correctness:** 4
**Technical Novelty And Significance:** 4
**Empirical Novelty And Significance:** 4
**Recommendation:** 8

**Clarity, Quality, Novelty And Reproducibility:**

[Clarity & Quality & Novelty]
+ The paper is well-written and easy to follow.
+ The analysis and proposed loss have high contribution and novelty.
- The source code is not currently available.


**Strength And Weaknesses:**

[Strenght]
+ the analysis and finding of regression's reformulation as a classification problem.
+ propose an ordinal entropy loss to learn high-entropy feature representations which preserve ordinality
+ significantly improved performance on various datasets and tasks.


**Summary Of The Paper:**

This paper presents an analysis of regression's reformulation as a classification problem and proposes an ordinal entropy loss to learn high-entropy feature representations which preserve ordinality based on the analysis.
The proposed method outperforms the previous regression loss, classification loss, and other comparison methods in various benchmark datasets. A thorough ablation study also properly supports the author's claim.



**Summary Of The Review:**

Based on the strength, weaknesses, and novelty, I think this paper is novel and has high contributions. But I am not absolutely certain about my decision, I will adjust my decision after checking other reviewers' opinions.

---

> ### Author Response · Authors · 2022-11-19
> **Thanks for the careful review and constructive suggestions**
>
> Thank you for the careful review and constructive suggestions.  We are happy that the novelty and contribution of our work are recognized.

---

### Official Review · Reviewer_1mqu · 2022-10-25

**Confidence:** 4
**Correctness:** 2
**Technical Novelty And Significance:** 2
**Empirical Novelty And Significance:** 2
**Recommendation:** 3

**Clarity, Quality, Novelty And Reproducibility:**

There are some clarity issues, like an unfinished sentence in the second paragraph of section 3.1. In Lemma 1, even though the statement mentioned \eta as consideration, it does not appear in the stated inequality. The authors are recommended to present it such that it is clear to the reader.

**Strength And Weaknesses:**

**Strengths**:
- The authors aim to provide an explanation of the phenomenon that discretizing continuous targets into classification often time performs better than optimizing the original loss.
- The proposed regularizer is motivated by the ordinal entropy.
- The author shows the performance benefit of the regularizer compared to just optimizing the MSE loss.

**Weakness**:

Even though the authors aim to provide an explanation of why converting continuous tasks into a classification is better than running a regression, I don't think the provided analysis is sufficient enough to establish the claim. First, the analysis that the regression objective only optimizes H(Z|Y) is an interpretation/approximation under certain assumptions. The authors establish the relation, which says that the MSE is a proxy for maximizing H(Z|Y). However, the analysis cannot say for sure that the MSE also optimizes something else (like a proxy to H(Z)) as the relations are not exact. In addition, the analysis is only conducted on a linear regressor, not a very non-linear deep network. An analysis that works for linear regressor cases does not always translate to deep neural network settings. Therefore, a more thorough theoretical and empirical analysis needs to be conducted to establish the such claim.

In the experiment, the authors did not compare the results with classification-based networks. This is important to back up the claim made by the authors that by modifying the continuous loss, the regression objective can outperform the classification baselines.

**Summary Of The Paper:**

The authors analyze multiple deep learning prediction cases with continuous targets where discretizing the target into a classification problem results in better results compared to directly employing regression loss functions. The authors argue that the root cause of the problem is that the regression objective only optimizes the conditional entropy H(Z|Y) but not the marginal entropy H(Z), whereas the cross-entropy objective optimizes both. The authors then propose a remedy to the situation by incorporating a regularizer to also maximize H(Z). The author finally demonstrates that the regularized improve the performance of the regression model.

**Summary Of The Review:**

The paper contains some weaknesses in the claim.
Therefore, I recommend rejection.

---

> ### Author Response · Authors · 2022-11-19
> **Thanks and response to concerns**
>
> Thank you for the careful review and constructive suggestions.
>
> For the specific questions:
>
> ---
>
> **Q1: Regression objective only optimizes $\mathcal H(Z|Y)$ is an interpretation or approximation under certain assumptions.**
>
> **A1:** Thanks for your feedback.  Making assumptions is often necessary to advance any form of analysis.
>
> We make two assumptions and justify them as follows:
>
> - $Z^c|Y$ follows a Gaussian distribution - this assumption is standard in the literature when analyzing features [1,2] and entropy [3].
> - a fixed $\cos {\phi_i}$ - at each iteration of training, $\cos {\phi_i}$ is a constant value.
>
> [1] Yang et al. Free Lunch for Few-shot Learning: Distribution Calibration. ICLR. 2021.
>
> [2] ​​Salakhutdinov et al. One-shot learning with a hierarchical nonparametric bayesian model. In ICML workshop, 2012.
>
> [3] Misra et al. Estimation of the entropy of a multivariate normal distribution. Journal of multivariate analysis. 2005.
>
> ---
>
> **Q2: The analysis cannot say for sure that the MSE also optimizes something else (e.g. a proxy to $\mathcal H(Z)$) as the relations are not exact.**
>
> **A2:** Thanks for pointing out this ambiguity.  You are right that the MSE loss is proportional to $H(Z|Y)$ with an additive term:
>
>
> > $L_{\text{mse}} \propto \mathcal H(Z|Y) + \mathcal D_{KL}(Z||Z^c |Y)$
>
> The additive term is the KL divergence between the two conditional distributions $P(Z|Y)$ and $P(Z^c|Y)$, where $Z^c_i$ are feature centers of Z.  Minimizing this divergence will either force $Z$ closer to the centers $Z^c$, or move the centers $Z^c$. By definition, however, the cluster centers $Z^{c}_i$ cannot expand beyond $Z$’s coverage, so features Z must shrink to minimize the divergence. As such, the entropy $ \mathcal H(Z)$ will not be increased by this term.
>
> We have modified Theorem 1 to be more specific in stating that “minimizing $L_{\text{mse}}$ is a proxy for minimizing $\mathcal H(Z|Y)$ without increasing $\mathcal  H(Z)$”.
>
> ---
>
> **Q3: The analysis is only conducted on a linear regressor and not a non-linear deep network. An analysis that works for linear regressor cases does not always translate to deep neural network settings.**
>
> **A3:** We model the network as an encoder $\phi (X)$ on input $X$ plus a linear regressor $f$ on feature $Z$, i.e.
>
> >$Y = f(Z), ~ \text{where}~~ Z = \phi(X)$.
>
> It is common to use linear regressors on the learned feature Z [4, 5].  As we do not make any assumptions on $Z$ and $\phi$ can be arbitrarily non-linear and/or deep.  While the analysis can be extended beyond the given $Z$, it does not invalidate our current findings.  We leave the extension as future work.
>
> [4] Sun et al. Integral Human Pose Regression. ECCV. 2018.
>
> [5] Goodfellow et al. Deep Learning. MIT press. 2016.
>
> ---
>
> **Q4: Compare with Classification-based models to back up the claim that the regression objective can outperform the classification baselines with the proposed ordinal entropy.**
>
> **A4:** We do not claim that we outperform classification baselines.
>
> Our work is predicated on the observation that classification outperforms regression.  We propose ordinal entropy terms to improve regression, but these improvements do not outperform classification.
>
> We have revised our text and Table 3 to state this point more explicitly.
>
> ---
>
> **Q5: In Lemma 1, even though the statement mentioned $\eta$ as consideration, it does not appear in the stated inequality**
>
> **A5:** Thanks for pointing this out.  $\eta$ is exploited in the detailed proof, which can be found in Appendix A.  We have modified the main text to include it in the inequality.

---

> > ### Comment · Reviewer_1mqu · 2022-12-09
> > **Thanks for the feedback**
> >
> > Dear authors.
> >
> > Thanks for the feedback. I have read it and other reviewer’s comments.
> >
> > My main concern about the paper is the flaw in the arguments presented in the paper. The analysis in the paper is built on several assumptions, e.g., the use of linear regressor and other assumptions in Lemma 1.
> >
> > Since the analysis is based on the approximations of the original problem, we have to be careful in making claims. Stating a positive claim under the approximation is OK as long as the condition is mentioned. However, for stating a negative claim under the approximation, we have to be extra careful.
> >
> > We cannot simply say the MSE loss for deep networks does not improve H(Z), just because the approximate model does not seems to improve H(Z). The issue could be on how we develop our assumptions. Under different assumptions, they could behave differently. The original problem statements could also behave differently.
> >
> > For my concern about the simplification to the linear predictor, the authors stated that this simplification is common in the literature. I agree with that statement for older literature. However, there are some recent developments in deep learning theory that consider more complexity of the deep network, e.g., using deep linear networks or neural tangent kernels (see [1] and [2]). I think for analyzing the problem stated in the paper, more complex tools need to be used, as previous studies discovered that the characteristics of deep networks are different from the ones in the linear predictor.
> >
> > Therefore, I stick with my original recommendation.
> >
> > [1] Roberts, D. A., Yaida, S., & Hanin, B. (2022). The Principles of Deep Learning Theory: An Effective Theory Approach to Understanding Neural Networks. Cambridge University Press.
> >
> > [2] Arora, Raman, Sanjev Arora, Joan Bruna, Nadav Cohen, Simon Du, Rong Ge, Suriya Gunasekar, et al. "Theory of deep learning." (2019).

---

> > > ### Author Response · Authors · 2022-12-11
> > > **Thanks for the feedback and raising several valuable points**
> > >
> > > Dear Reviewer 1mqu,
> > >
> > > Thank you for the careful review and constructive suggestions.
> > >
> > > For the specific concerns:
> > >
> > > ---
> > >
> > > **Q1: Assumptions in Lemma 1**
> > >
> > > A1: In Lemma 1, we only assume the maximum bin width $\eta$ is small enough: Regression is the extreme case where $\eta$ is equal to 0.
> > >
> > > ---
> > >
> > > **Q2: The analysis is built on several assumptions, and regression with MSE could behave differently under different assumptions.**
> > >
> > > A2: Thanks for pointing this out. While different assumptions may result in different conclusions, reasonable assumptions are acceptable and commonly will not result in a wrong conclusion. We have justified the reasonability of all the assumptions as discussed above, and the conclusion is also grounded by experimentation and visualization. Although a more general analysis without assumptions is preferred, making assumptions is often necessary to advance any form of analysis. Besides, it is worth mentioning that our theoretical analysis is the first foray into understanding why classification commonly outperforms regression in computer vision.
> > >
> > > ---
> > >
> > > **Q3: There are some recent developments in deep learning theory that consider more complexity of the deep network, e.g., using deep linear networks or neural tangent kernels as the predictor.**
> > >
> > > A3: While it is true that some recent methods exploit advanced predictors rather than linear ones, it does not invalidate our current findings. Besides, it is still common to exploit linear predictors nowadays, like a convolution layer with kernel size equal to $1 \times 1$ or $3 \times 3$. While the analysis can be extended beyond the linear predictor, we leave the extension as future work. Intuitively, the linear/nonlinear predictors only influence the mapping between the feature space and the label space. Thus, our conclusion still holds as long as we can eliminate the influence of the nonlinear predictor mapping. Besides, our conclusion also holds for those nonlinear predictors, which consist of a linear regressor followed by a nonlinear activation function, as long as the nonlinear activation function does not break the proportional relationship in Theorem 1.
> > >
> > > ---
> > > **Q4: Stating a negative claim under the approximation should be extra careful.**
> > >
> > > A4: Thanks for pointing this out. Here we would like to justify why the approximations (2 in total) can not influence the existence of $\mathcal H(Z)$ as follows:
> > >
> > > - Approximate $y_i$ with bin centers $z_i$ based on Lemma 1:  This approximation will introduce the discretization error, yet, the error can be made arbitrarily small if the bin width $\eta$ is sufficiently fine. In other words, even if $\mathcal H(Z)$ exists in the approximation, it can be made arbitrarily small.
> > >
> > > - The $\propto$ approximation in Theorem 1, i.e. $L_{\text{mse}} \propto \mathcal H(Z|Y) + \mathcal D_{KL}(Z||Z^c |Y)$ : If the $\mathcal H(Z)$ exist in $L_{\text{mse}} $ => the $L_{\text{mse}} $ can be increased or decreased only by $\mathcal H(Z)$ while the right side do not contain $\mathcal H(Z)$ and will not change => the $\propto$ relationship can not hold =>  $\mathcal H(Z)$ dose not exist in $L_{\text{mse}} $ .
> > >
> > > Thanks.
> > >
> > > &nbsp;
> > >
> > > Yours sincerely,
> > >
> > > Authors

---

> > > > ### Comment · Reviewer_1mqu · 2022-12-11
> > > > **Thanks for the feedback**
> > > >
> > > > What I mean by the approximations are not the approximations in the equation. Rather, I refer to approximating the original problem, i.e., optimizing MSE loss in deep networks with another model.

---

> > > > > ### Author Response · Authors · 2022-12-12
> > > > > **Thanks for the feedback**
> > > > >
> > > > > Dear Reviewer 1mqu,
> > > > >
> > > > > Regarding concerns about approximating with models or assumptions, we hope our responses to Q2 and Q3 have addressed these concerns. The responses to Q4 answer the potential questions about why approximating MSE loss with  $\mathcal H(Z|Y) + \mathcal D_{KL}(Z||Z^c |Y)$ can not influence the existence of $\mathcal H(Z)$.
> > > > >
> > > > > Thanks.
> > > > >
> > > > > &nbsp;
> > > > >
> > > > > Yours sincerely,
> > > > >
> > > > > Authors

---

### Official Review · Reviewer_NgkU · 2022-10-25

**Confidence:** 3
**Correctness:** 4
**Technical Novelty And Significance:** 4
**Empirical Novelty And Significance:** 3
**Recommendation:** 8

**Clarity, Quality, Novelty And Reproducibility:**

Very good.
Clear, of high quality and mostly reproducible.

**Strength And Weaknesses:**

Strengths
-Well written.
-Technically sound.
-Analytical and intuitively well motivated.
-Novel contributions to deep regression, of wide application.
-Sound proposed methodology.


Weaknesses
-minor typos. For instance, "The simplest mapping The feature encoding..."
-complexity/efficiency discussion. Since the proposed loss term seems to be quadratic in the number of observations, it would be interesting to discuss the efficiency or the impact on the batch size.
-in ordinal regression (also known as ordinal classification), there are works adding a regression component to the classification loss. Although quite different from the current proposal, it may be interesting to comment and put into perspective in the related work.
https://arxiv.org/abs/2202.05167
https://aclanthology.org/2022.coling-1.407/




**Summary Of The Paper:**

The paper proposed to loss term for deep regression models to increase the entropy of the learned representation and with that increase the performance of the predictive model. An additional term is also proposed to decrease the entropy of the class conditioned representation.

**Summary Of The Review:**

see above

---

> ### Author Response · Authors · 2022-11-19
> **Thanks and response to concerns**
>
> Thank you for the careful review and constructive suggestions.
>
> For the specific questions:
>
> ---
>
> **Q1: The efficiency or impact of sample size (M)**
>
> **A1:** Thanks for pointing this out. Please see the general response G1.
>
> ---
>
> **Q2: Related works on ordinal regression.**
>
> **A2:** Thanks for pointing this out. Please see the general response G3.

---

### Author Response · Authors · 2022-11-19
**Thanks for the reviews, answers to general questions and summary of key paper changes**

We thank the reviewers for their careful and thoughtful reviews. R1, R3 and R4 all appreciated our work’s novelty and contribution as well as the technically sound theoretical and empirical demonstrations.

We address three general concerns here and offer detailed responses individually to each review.

---

**G1 (R1, R4): Is the proposed regularizer defined on the mini-batch or the entire dataset? What is the efficiency or impact on the number of observations?**

**A1:** Thanks for pointing out this ambiguity.  We have clarified this for the revision.

$L_d$ : entropy regularizer

$L_d$  is defined over M samples, where M is a (subsampled) number of feature centers $z_{c_i}$ in the mini-batch.  Note that the number of feature centers is determined by the number of different output targets.

Figure 4a in the paper shows the gradual decrease and then increase of MSE and variance with the increase in M (see explanation in Section 5.4).

The computational complexity of $L_d$ is quadratic with respect to M and does add compute time for training.  The synthetic experiments (Table 1) use a simple 2-layer MLP, so adding $L_d$ has a significant relative increase in time with large M.  For depth estimation, which uses a larger ResNet-50 backbone, the relative increase in time and memory are much less (27% and 0.3%, respectively, see Table 2), despite a large M=3536, where 3536 is a 16x subsampling of the total number of pixels in an image.  Note that the increases are only for training and do not add compute demands for inference.

$L_t$ : tightness regularizer

$L_t$ is defined over $N_b$ samples, where $N_b$ is the batch size. Again, the added time and memory with $L_t$ is also negligible (0.08% and ~0%, respectively), even with M=3536 (Table 2).

Table 1.
|            M             | Training time (s) | Memory taken (MB) |
|:------------------------:|:-----------------:|:-----------------:|
|            0             |        155        |       2163        |
|            10            |        296        |       2163        |
|           100            |        327        |       2163        |
|           1000           |       1033        |       2205        |

Table 2.
|            M             |  Regularizer   | Training time (s) | Memory taken (MB) |
|:------------------------:|:--------------:|:-----------------:|:-----------------:|
|            0             | no regularizer |       1836        |       12363       |
|           100            |   $L_d+L_t$    |       1877        |       12379       |
|           1000           |   $L_d+L_t$    |       1999        |       12395       |
| 3536 |     $L_d$      |       2342        |       12405       |
| 3536 |   $L_d+L_t$    |       2344        |       12405       |

---

**G2 (R3, R4): Release the Code**

**A2:** We have submitted the code as supplementary materials.

---

**G3 (R1, R4): Related works on ordinal regression:**

**A3:** Thanks for pointing these out. We added a discussion to the related works.

[Castagnos’22, Gong’22, Polat’22] and our work all aim to preserve ordinality via label distances. [Castagnos’22, Polat’22] exploit ordinality to improve classification performance. [Gong’22] exploits ordinality for imbalanced regression. The main goal of our work is to encourage a higher-entropy feature space. We preserve ordinality to regularize random increases in entropy.

---

Key changes to our paper are summarized as follows:

- Experiments on the influence of $\lambda_t$ in Appendix E, as requested by Reviewer 4 (eTrs).
- Experiments on the efficiency vs. sample size M in Appendix F, as requested by Reviewer 4 (eTrs) and Reviewer 1 (NgkU).
- Clarification on theorem 1, as requested by Reviewer 2 (1mqu).
- Discussion on ordinal regression in Related Works, as requested by Reviewer NgkU and Reviewer 4 (eTrs).
- Typos.

---

### Author Response · Authors · 2022-12-11
**Thanks for taking the time to review and discuss our paper.**

Dear Reviewers, dear AC,

We thank you for taking the time to review and discuss our paper.  As we have not heard back from reviewers eTrs, ejHX, and NgkU, we hope that this means that our paper is regarded positively, especially as they have already pointed out that our paper is novel and can be a contribution to the ML community.

We also thank reviewer 1mqu for engaging in discussion and raising several valuable points and we will take these into consideration when revising our paper.  However, we would like to raise a few final points, which we hope everyone will take into consideration when evaluating our paper:

- use of assumptions: We think it is unreasonable to expect not to make any assumptions for an analysis paper such as ours.  And indeed, different assumptions may lead to different results.  We take care to justify each of our assumptions and ground our conclusions with experimentation and visualization.

- factoring more complexity of the network, like neural tangent kernels [1] and deep linear networks, for analysis. These works are interesting and we will consider them for future work.  Nevertheless, these developments do not invalidate our work.  Our analysis is the first foray into comparing the differences between classification and regression in learning for deep neural networks.

[1] Jacot et al. Neural tangent kernel: Convergence and generalization in neural networks. NeurIPS. 2018

Thank you again to all the reviewers and AC.

&nbsp;


Yours sincerely,

Authors

---

### Decision · Program_Chairs · 2023-01-20

**Decision:**

Accept: poster

**Justification For Why Not Higher Score:**

The paper claims are interesting but limited to linear network settings

**Justification For Why Not Lower Score:**

Analyzing why classification method outperforms regression method is an interesting topic.

**Metareview: Summary, Strengths And Weaknesses:**

The paper gives an explanation on why regression model underperform with respect to classification based regression method from an information theory point of view and propose a regularizer based on ordinal entropy  that improves the regression with MSE loss.

The paper was discussed between authors reviewers and AC.

The main strength of the paper is in analyzing under certain assumptions the performance of MSE loss. The main weakness of the work is that the analysis is limited to linear models and to restricting assumptions and does not explain what happens in the deep setting and beyond the assumptions listed in the paper.

Overall the reviewers had positive views on the work with the exception of reviewrer 1mqu. We encourage the authors to be more precise in listing the contributions of the work and to take into account the constructive  feedback of Reviewer 1mqu on better scoping the work and to not extrapolate the results to other settings without any proof.

**Note From Pc:**

if the above contains the word "oral" or "spotlight" please see: "oral" presentation means -> notable-top-5% and "spotlight" means -> notable-top-25%. As stated in our emails, we are disassociating presentation type from AC recommendations

**Summary Of Ac-Reviewer Meeting:**

N/A